# A Learning-Driven Multi-Agent Approach to Metadata Extraction from Legal Contracts

## Abstract

High-quality metadata is essential for downstream legal retrieval applications, yet its generation remains challenging due to the need to balance accuracy, scalability, and computational cost. We present a reinforcement learning (RL)–based orchestration framework that coordinates specialized LLM agents to optimize the cost–quality trade-off in contract metadata extraction. The orchestrator is trained with a goal-conditioned reward balancing field-level extraction quality against token usage, and instantiated with two learners: a goal-conditioned Deep Q-Network (DQN) and Proximal Policy Optimization (PPO). We evaluate performance in two settings of varying complexity: (1) a uniform enterprise contract-summary dataset and (2) full-text CUAD contracts. On CUAD, a quality-first PPO policy improves average quality by about 2% over an agentic LLM with rigid orchestration, while an efficiency-first policy reduces token usage by 35% with only a modest quality trade-off. On enterprise summaries, DQN achieves 65% token savings at a 28% quality trade-off, making it a strong option under strict budget constraints. We further introduce a multi-pass variant that preserves baseline quality while substantially reducing cost. Overall, RL-based orchestration offers a practical path toward scalable and economically sustainable legal metadata generation.

## 1 Introduction

The rapid growth of unstructured legal and regulatory text makes reliable, scalable metadata extraction a central NLP challenge (Chalkidis et al., 2020). High-quality *metadata* (structure, jurisdiction, parties, statutory context) underpins FAIR principles and directly affects retrieval quality and legal research efficiency; conversely, inconsistent metadata creates bottlenecks in discovery, compliance, and contract analysis (Wilkinson et al., 2016; Premkumar, 2025).

Classical solutions span (i) **manual curation** by experts (accurate but expensive) (Graham et al., 2025); (ii) **NLP pipelines** (NER, rules, supervised models) (Yuan & Zhang, 2021; Chalkidis et al., 2020); and (iii) **ontology/heuristic methods** that normalize labels and relations (Hoekstra et al., 2007; Ren et al., 2022; Wilkinson et al., 2016). While LLMs and resources like LegalBERT and CUAD have advanced extraction (Chalkidis et al., 2020; Hendrycks et al., 2021), keyword-style retrieval remains shallow (Sleimi et al., 2021) and single-pass prompting is costly and brittle. A framework that *explicitly balances metadata quality and compute* is still needed.

We address this with an RL-based *orchestration* framework that learns when and how to invoke specialized LLM agents. A goal-conditioned reward trades off field-level quality against token usage, producing a family of policies along a cost–quality Pareto frontier. We instantiate two learners: a goal-conditioned DQN with a two-phase curriculum (Mnih et al., 2015) and PPO (Schulman et al., 2017).

**Contributions.** (i) An RL orchestration framework for legal metadata extraction with programmable quality–cost rewards; (ii) a comparative study of goal-conditioned DQN vs. PPO and a *Multi-Pass* workflow for cost-aware extraction; (iii) evaluations on CUAD and an enterprise contract-summary corpus with ablations and Pareto analysis; and (iv) a practical blueprint for reproducible, deployable document-understanding pipelines.

## 2    RELATED WORK

**Metadata extraction in law.** Classical pipelines for contract understanding combine named-entity recognition (parties, dates, amounts) and rule-based post-processing for normalization and validation (Chalkidis et al., 2020; Yuan & Zhang, 2021). Ontology-driven methods provide schema alignment and controlled vocabularies to improve downstream interoperability and de-duplication (Hoekstra et al., 2007; Ren et al., 2022; Wilkinson et al., 2016). The CUAD dataset (Hendrycks et al., 2021) catalyzed systematic evaluation of clause-level understanding and remains a challenging benchmark for long, heterogeneous legal contracts.

**LLM agents and document ETL.** Agentic frameworks decompose document tasks into tool-augmented steps such as retrieval, parsing, normalization, and validation. DocETL formalizes multi-step document extraction, evaluation, and lineage (Shankar et al., 2024), while LAW scales agentic workflows to large regulatory corpora (Watson et al., 2024). Domain systems such as LegalMind experiment with RL-style or GRPO-style alignment for legal decision support (Vara et al., 2025). These works demonstrate that *workflow design* (which tool to call, when to retry/stop) is as important as model capability—yet most systems do not explicitly optimize token cost or reason about a *global* cost–quality objective across steps.

**RL for orchestration and alignment.** RLHF/DPO align LLMs to preferences (Rafailov et al., 2023). For policy optimization, PPO is a standard, stable on-policy method with strong empirical performance (Schulman et al., 2017); CoPPO extends PPO to cooperative multi-agent settings (Wu et al., 2021). IMPALA provides a high-throughput actor–learner architecture for large-scale RL (Espeholt et al., 2018). Programmable reward methods (e.g., GRPO) reduce human feedback but require careful shaping to prevent reward hacking (Vara et al., 2024; Mroueh et al., 2025). Our work targets the *orchestration* layer: the policy decides which specialist agent to invoke (or to stop) to maximize a quality–cost reward over a document episode.

COMPARATIVE VIEW OF RL FAMILIES (TABLE 1)

Table 1 summarizes canonical RL families and their trade-offs for document orchestration. We highlight the implications for our problem:

- **Policy-gradient and actor–critic (REINFORCE/A2C/PPO).** These methods directly optimize stochastic policies and cope well with non-differentiable rewards and action masking. PPO's clipped objective stabilizes learning under shifting reward scales (e.g., changing token prices or goal weights), which matches our non-stationary cost–quality setting (Williams, 1992; Konda & Tsitsiklis, 2000; Mnih et al., 2016; Schulman et al., 2017).

- **Value-based (DQN).** With a small, discrete action set (select specialist ∪ STOP), DQN is sample-efficient and excels at learning sharp cost-saving behaviors, especially when coupled with action masking and coverage shaping. Goal-conditioning allows one network to realize a *family* of policies along the Pareto frontier (Mnih et al., 2015).

- **Cooperative PPO (CoPPO).** Useful when multiple agents must act *simultaneously* with communication costs; our orchestration is *sequential* with one action per step, so full multi-agent credit assignment is unnecessary overhead (Wu et al., 2021).

- **Bandits.** Excellent for single-shot choices (e.g., pick *one* agent once), but our task requires temporal credit assignment (decide *which* agent, *in what order*, and *when to stop*), which bandits cannot capture (Langford & Zhang, 2007).

- **Hierarchical RL.** Powerful for long-horizon problems with latent subgoals (Barto & Mahadevan, 2003; Nachum et al., 2018), but introduces extra complexity (skills/options) that is not essential when the action space already reflects high-level skills (specialist agents).

- **Model-based RL (DreamerV3).** Sample-efficient and appealing for risky domains (Hafner et al., 2023), but the environment here is dominated by black-box LLM calls; learning a faithful world model of tokenized API behavior is non-trivial and offers limited payoff compared to direct policy optimization.

- **Offline RL/Imitation.** Attractive when exploration is risky or expensive (Levine et al., 2020), yet our objective explicitly values *adaptive* querying (quality vs. cost) that benefits from online signals (confidence, residual fields).

- **IMPALA.** Valuable when scaling to thousands of actors (Espeholt et al., 2018); our training is modestly sized and API-latency-bound, so simpler learners suffice.

**Why PPO and goal-conditioned DQN suit orchestration.** (i) **Discrete, masked actions.** Choosing among a handful of specialist agents plus STOP is naturally handled by DQN or PPO with action masking (Table 1). (ii) **Non-stationary costs and goals.** Token prices and target quality thresholds vary across deployments; PPO's clipping and advantage normalization, and DQN's reward clipping and double targets, provide robustness when $(\lambda, \mu)$ and document lengths shift. (iii) **Pareto control.** Goal conditioning lets a single network realize quality-first *and* efficiency-first behaviors; practitioners simply dial the goal vector at inference (Section 3). (iv) **Stopping.** Both methods handle a learned STOP action, which is crucial to avoid over-querying agents and to respect budget constraints. (v) **Simplicity and compute.** Compared to hierarchical or model-based approaches, PPO/DQN deliver strong performance with minimal infrastructure—important for enterprise deployments where the environment step is dominated by LLM API latency rather than simulator throughput.

In short, PPO provides stable, high-quality policies for complex contracts (e.g., CUAD), while goal-conditioned DQN achieves sharp token savings on more uniform enterprise summaries—exactly mirroring the empirical trends we report in Section 3 and the Results.

## 3 PROPOSED FRAMEWORK

### 3.1 PROBLEM AND DATASETS

We study extraction of a schema $\mathcal{F}$ of metadata fields (e.g., *party*, *effective_date*, *governing_law*) from two corpora: (i) CUAD (Hendrycks et al., 2021), sampled to 264 train and 66 test contracts, and (ii) an enterprise corpus of telecom supplier contracts spanning 15 years (2,268 train, 567 test) derived from agreement summaries. Contracts vary widely in length and structure; only a small fraction contains labelled clauses.

### 3.2 MULTI-AGENT ORCHESTRATION

We formalize the problem as sequential decision-making. The environment state $s_t$ encodes (i) which fields are filled, (ii) document features (e.g., amendment/financial/length), and (iii) cumulative token usage. The action space $\mathcal{A}$ consists of calls to specialist agents—regex+LLM extractors (e.g., `PartiesAgent`, `DatesAgent`, `GoverningLawAgent`)—and a STOP action. Each action consumes tokens and yields a value + confidence. A fast retrieval step selects relevant chunks per field, reducing context length.

### 3.3 REWARD AND GOAL CONDITIONING

We cast orchestration as a sequential decision process with per-step reward that trades off metadata quality against compute. Let $\mathcal{F}$ be the set of fields (e.g., *party*, *effective_date*, *governing_law*). At step $t$, the system maintains the best predictions $\hat{y}_f^{(t)}$ and confidences $c_f^{(t)} \in [0, 1]$ for each $f \in \mathcal{F}$. We define a normalized, field-aware quality score

$$Q_t = \frac{1}{Z} \sum_{f \in \mathcal{F}} w_f \underbrace{s_f(\hat{y}_f^{(t)}, y_f^\star)}_{\text{exact/partial match}} \quad \text{with} \quad Z = \sum_f w_f,$$

where $y_f^\star$ is the ground truth (available during training), $w_f$ are importance weights, and $s_f(\cdot)$ gives partial credit (e.g., token-level F1 for spans, type/format checks for dates and amounts, and Jaccard overlap or Levenshtein similarity for strings). Quality improvement is $\Delta Q_t = Q_t - Q_{t-1}$.

We charge cost at the same granularity. For an action $a_t$ (calling a specialist agent or STOP), we record prompt and completion tokens $(\tau^{\text{pr}}, \tau^{\text{cm}})$ and define

$$\Delta \text{Tokens}_t = \alpha \tau^{\text{pr}} + \beta \tau^{\text{cm}} + \kappa \, \texttt{tool\_io} + \xi \, \texttt{invalid}.$$

Table 1: RL models and typical applications relevant to multi-agent orchestration.

| Algorithm / Class | Pros | Cons | Example Use Cases | Key References |
|---|---|---|---|---|
| **Policy Gradient** (REINFORCE), Actor–Critic (A2C), **PPO** | Foundational RL family; scalable to complex tasks; widely used in RLHF. | REINFORCE high variance; Actor–Critic/PPO more complex to implement. | Robotics, strategy games, orchestration, LLM fine-tuning. | (Williams, 1992; Konda & Tsitsiklis, 2000; Schulman et al., 2017; Mnih et al., 2016; Zhang et al., 2022) |
| **CoPPO** | Better stability in cooperative settings. | Communication overhead limits scaling. | Multi-agent control. | (Wu et al., 2021) |
| **DQN** | Stable with replay/targets; strong in discrete tasks. | Discrete-only; long-horizon issues. | Atari, recommenders. | (Mnih et al., 2015) |
| **IMPALA** | Very high throughput. | Infra-hungry at small scales. | Massive training regimes. | (Espeholt et al., 2018) |
| **GRPO / MLPO** | Low/no human labels; transparent shaping. | Reward hacking risk. | LLM alignment and tools. | (Vara et al., 2024; Mroueh et al., 2025) |
| **Bandits** | Fast, simple. | No temporal credit. | One-shot selection. | (Langford & Zhang, 2007) |
| **Hierarchical RL** | Long-horizon planning. | Implementation complexity. | Multi-stage workflows. | (Barto & Mahadevan, 2003; Nachum et al., 2018; Estornell et al., 2025) |
| **Model-Based (DreamerV3)** | Sample efficient. | Model bias sensitivity. | Safety-critical domains. | (Hafner et al., 2023) |
| **Offline / Imitation** | No online exploration. | Distribution shift risk. | Regulated domains. | (Levine et al., 2020) |

where $\alpha, \beta$ weight prompt vs. completion, $\kappa$ captures fixed per-call overhead (e.g., retrieval or post-processing), and $\xi$ penalizes invalid actions (e.g., selecting an agent for an already-filled field when action masking is disabled).

The per-step reward is

$$R_t = \lambda \Delta Q_t - \mu \Delta \text{Tokens}_t + \eta \Delta C_t + b_t, \tag{1}$$

where $\eta \Delta C_t$ optionally rewards *coverage* improvements (e.g., number of newly filled fields or increased confidence $c_f^{(t)}$), and $b_t$ is a terminal bonus/penalty:

$$b_t = \begin{cases} +\rho & \text{if } a_t = \text{STOP and } Q_t \geq \gamma_g, \\ -\rho' & \text{if } a_t = \text{STOP and } Q_t < \gamma_g, \\ 0 & \text{otherwise.} \end{cases}$$

Here $\gamma_g$ is a target quality threshold tied to the goal (see below). We use discount $\gamma = 0.99$ during learning and clip rewards to a small range to stabilize training.

**Goal conditioning.** Rather than training separate policies for "quality-first" and "efficiency-first" behavior, we *condition* both the policy and value functions on a goal vector $g$:

$$g = \big[\, \gamma_g,\ \lambda,\ \mu \,\big] \in [0, 1]^3,$$

where $\gamma_g$ is a desired quality target and $(\lambda, \mu)$ are the reward weights in equation 1. The goal $g$ is concatenated to the state representation (document features, remaining-fields mask, cumulative tokens, recent agent history). During training we sample $g$ from a curriculum (e.g., $\gamma_g \in \{0.70, 0.80, 0.90\}$ with $(\lambda, \mu)$ drawn from a simplex), which teaches a single network to realize a *family* of policies along the cost–quality Pareto frontier. At inference time, practitioners select $g$ to "dial" the behavior: higher $\gamma_g$ or $\lambda/\mu$ yields quality-centric behavior; lower $\gamma_g$ or higher $\mu$ yields efficiency-centric behavior.

**Shaping and safeguards.** To reduce myopic behaviors we add: (i) a small entropy bonus (via the RL losses below) to preserve exploration among similarly performing agents; (ii) a patience penalty if $\Delta Q_t \leq 0$ over $K$ consecutive steps (nudging STOP); and (iii) action masking that disallows agents for already-resolved fields and forbids obviously incompatible actions (e.g., re-running a date parser on a validated ISO date) to improve sample efficiency without changing the optimal policy. All shaping terms are small (typically $\leq 5\%$ of the magnitude of $\lambda$) to avoid reward hacking.

### 3.4 POLICY LEARNING: DQN VS. PPO

**State and actions.** The state $s_t$ encodes: (a) a field-status vector (filled/empty, confidence, validation flags), (b) document features (length, header signals, clause heuristics), (c) cumulative tokens and recent per-agent costs, and (d) the goal $g$. The discrete action set contains one action per specialist agent plus STOP. We apply *feasibility masks* to zero out illegal actions before sampling/argmax.

**Goal-conditioned DQN (Mnih et al., 2015).** We learn $Q_\theta(s, a \,|\, g)$ with double Q-learning and prioritized replay:

- *Architecture.* A small MLP over $s \oplus g$ with optional dueling heads (value/advantage) to improve stability on sparse $\Delta Q_t$.
- *Targets.* $y = r + \gamma \, Q_{\bar{\theta}}\big(s', \arg\max_{a'} Q_\theta(s', a' \,|\, g)\big)$ with target network $\bar{\theta}$ updated every $K$ steps.
- *Replay.* Prioritized experience replay with $p_i \propto |\delta_i|^\alpha$ and importance weights $\beta$ annealed to 1. We store $(s, g, a, r, s', \text{mask})$, enabling off-policy reuse across many goals $g$ for the same transitions.
- *Loss.* Huber loss on TD error with gradient-norm clipping; reward clipping to $[-1, 1]$ improves robustness to token-spike outliers.
- *Exploration.* $\epsilon$-greedy with linear decay; masked actions are never sampled. We also add a tiny probability of sampling the STOP action early to learn when to terminate.

DQN is sample-efficient and excels when the action set is moderately sized and mostly discrete (our case). It tends to learn sharp cost-saving behaviors once the coverage/stop shaping is in place.

**Proximal Policy Optimization (PPO) (Schulman et al., 2017).** We learn a stochastic policy $\pi_\theta(a \,|\, s, g)$ and value $V_\phi(s, g)$ with clipped updates and generalized advantage estimation (GAE):

$$\mathcal{L}_{\text{PPO}} = \mathbb{E}\Big[\min\Big(r_t(\theta)\,\hat{A}_t,\ \text{clip}\big(r_t(\theta), 1-\epsilon, 1+\epsilon\big)\hat{A}_t\Big)\Big] - \beta_{\text{ent}}\,\mathcal{H}\big(\pi_\theta(\cdot|s, g)\big) + \beta_{\text{vf}}\big(V_\phi(s, g) - \hat{V}_t\big)^2,$$

where $r_t(\theta) = \frac{\pi_\theta(a_t|s_t, g)}{\pi_{\theta_{\text{old}}}(a_t|s_t, g)}$ and $\hat{A}_t$ uses $\text{GAE}(\lambda_{\text{GAE}})$ on the reward in equation 1. We apply:

- *Action masking.* Invalid actions receive probability zero before sampling; their logits are set to $-\infty$.

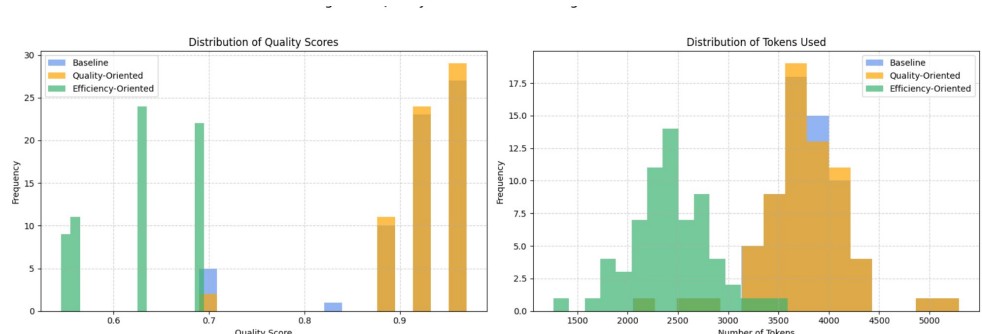

Figure 1: System architecture. The orchestrator observes the current state, selects a specialist agent or STOP, and receives a reward trading quality for token cost.

- *Early stopping.* Abort an epoch if the empirical KL exceeds a small threshold, avoiding policy collapse under highly goal-weighted rewards.
- *Normalization.* Running normalization for advantages and for scalar features (tokens, confidences) stabilizes training when switching between goals.
- *Curriculum over g.* Mini-batches mix goals; this teaches a single policy to realize both "quality-first" and "efficiency-first" behaviors. At inference, users only set $g$.

PPO is more stable than DQN under non-stationary reward scales (caused by changing $(\lambda, \mu)$ and document lengths) and typically attains slightly higher quality at modest extra token cost.

**Practical notes.** (i) We anneal $(\lambda, \mu)$ during training to sweep a range of Pareto points and later fix them at inference according to deployment constraints; (ii) the STOP action is critical—without a terminal bonus/penalty policies over-query agents; (iii) masking plus invalid-action penalties eliminates most wasted calls; and (iv) a tiny entropy bonus ($\beta_{\text{ent}} \in [10^{-4}, 10^{-3}]$) is enough to prevent premature collapse to a single favorite agent.

### 3.5 MULTI-PASS WORKFLOW

We introduce a two-pass strategy: a lightweight first pass (bandit/greedy) fills easy fields (dates, amounts, obvious parties), followed by PPO refinement on hard fields (governing law, clause categories). This reduces tokens while preserving accuracy and lends itself to human-in-the-loop triage.

### 3.6 IMPLEMENTATION DETAILS

We train in Python using API-served LLMs with frozen weights. Hyperparameters are tuned with Optuna (Akiba et al., 2019). For PPO on CUAD and enterprise data we adopt learning rate $3 \times 10^{-4}$, clip 0.2, and entropy coefficient 0.01; for DQN we use batch size 128, learning rate $10^{-4}$, and exploration decay over 2000 steps. We implement a resilient API call manager with adaptive rate limiting and retries to maintain throughput during large-scale evaluation.

## 4 EXPERIMENTAL SETUP

**Datasets.** We evaluate on CUAD (Hendrycks et al., 2021) and an enterprise dataset of telecom supplier contracts spanning 15 years (train/test partitions described below).

**Baselines.** (i) Single-pass prompted LLM; (ii) rule-based/regex pipeline; (iii) non-RL agent scheduling (static order).

**Metrics.** Field-level exact/partial match accuracy; macro-averaged quality; *tokens per document*; and *efficiency* (quality-per-token).

**Implementation.** We run orchestration on API LLMs (frozen weights). The proposed model was implemented using **Python 3** and **GPT-4**, with training executed on **Azure ML**

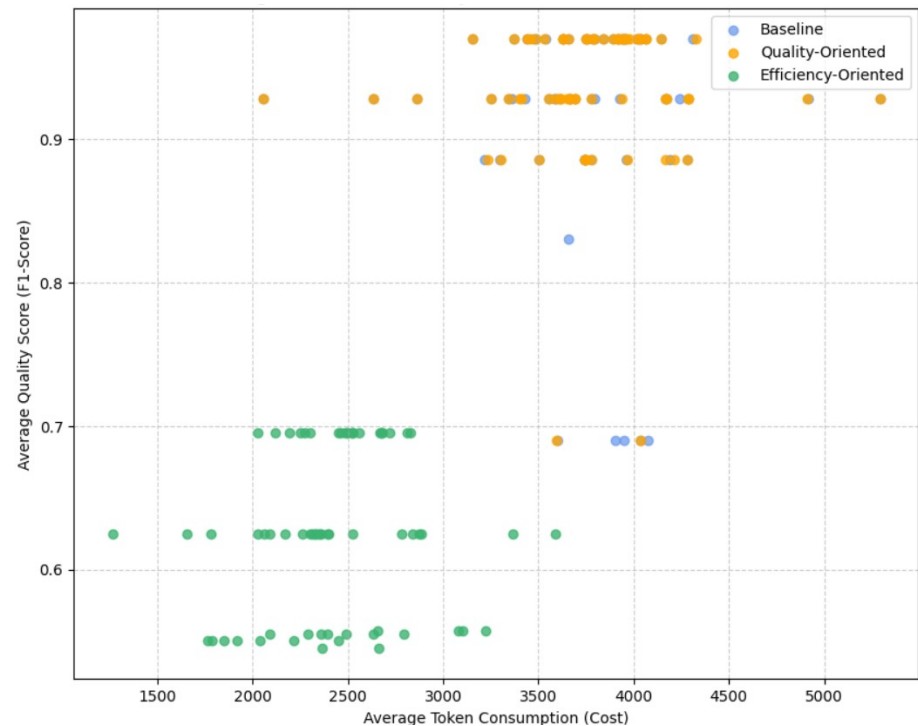

Figure 2: Quality vs. token usage across methods. Multi-Pass and efficiency-oriented policies approach the Pareto frontier.

`Standard_DS12_v2` virtual machines (4 vCPUs, 28 GB RAM, 56 GB disk). Training uses Adam with a cosine schedule; PPO clip = 0.2; DQN $\epsilon$-greedy decay; reward weights $(\lambda, \mu)$ swept on a validation split.

## 5 RESULTS

### 5.1 CUAD: QUALITY VS. COST

Table 2 reports results consistent with our draft: the quality-oriented policy improves average quality by ∼1.6% over baseline with negligible token change, while the efficiency-oriented policy reduces tokens by ∼35% at a moderate quality trade-off. Figure 2 shows clusters along the cost–quality frontier.

Table 2: CUAD policies: average quality (F1-style score) and average token consumption.

| Policy | Avg. Quality | Avg. Tokens |
|---|---|---|
| Baseline | 0.9163 | 3757.77 |
| Quality-Oriented | **0.9322** | 3762.36 |
| Efficiency-Oriented | 0.6264 | **2426.50** |

### 5.2 ENTERPRISE CONTRACTS

On enterprise summaries, DQN is especially effective for efficiency: the efficiency policy attains ∼72% of baseline quality with only ∼35% of tokens; PPO's efficiency policy yields ∼73% quality with ∼68% tokens. A Multi-Pass strategy recovers baseline quality at substantially lower cost by applying a cheap first pass then refining flagged fields.

### 5.3 Ablations and Analysis

**Reward weights.** Larger $\mu$ encourages earlier stopping and cheaper agent choices; larger $\lambda$ prioritizes difficult fields.

**Curriculum.** Goal-conditioned curricula accelerate DQN convergence on long contracts.

**Agent set.** Removing the party resolver causes the largest quality drop; removing the date parser increases tokens via re-tries.

**Decision traces.** Policies exhibit interpretable sequences: PPO defers to bandit on easy fields, then consults governing-law classification only when header confidence is low.

## 6 Discussion

**Which policy when?** PPO for maximal quality; DQN for strict budgets; Multi-Pass when both objectives matter. **Limitations.** Reward shaping can bias behavior; enterprise data may introduce sampling bias; multi-call orchestration adds latency. **Impact.** Orchestrated extraction improves compliance and discovery but requires care to avoid bias in metadata normalization.

## 7 Conclusion and Future Work

We presented an RL-based orchestration framework for contract metadata extraction that separates agent policy learning from LLM weights. Experiments on CUAD and enterprise corpora show quality improvements and substantial token savings. Future directions include multilingual corpora, richer state representations (e.g., LegalBERT embeddings), dynamic goal conditioning at inference, and enlarging the action space to select between LLMs per agent.

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
