# OpenReview forum: "A learning- driven multi agents approach to metadata extraction from legal contracts"
_ICLR.cc/2026/Conference — ICLR 2026 Conference Withdrawn Submission_

### Official Review · Reviewer_4qKx · 2025-10-21

**Soundness:** 2
**Presentation:** 3
**Contribution:** 2
**Rating:** 6
**Confidence:** 3

**Summary:**

This paper introduces a RL–based multi-agent orchestration framework for metadata extraction from legal contracts. The method coordinates specialized LLM-based agents  through an RL policy that optimizes the trade-off between extraction quality and computational cost . They explore 2 policy learners, it's goal-conditioned DQN and PPO, which is both conditioned on a goal vector controlling the balance between quality and efficiency. They further propose a multi-pass workflow, where a lightweight first pass handles easy fields before a refinement phase focuses on difficult ones. The paper demonstrates that RL-based orchestration offers a practical and scalable path to cost-aware legal metadata generation.

**Strengths:**

1. The paper addresses an underexplored but practically significant problem: learning to orchestrate multiple LLM agents for document understanding under explicit cost constraints.
2. The integration of RL decision-making with LLM-based document ETL pipelines is conceptually novel and bridges reinforcement learning with legal NLP.
3. Methodology is clearly presented, including formal definitions of state, actions, rewards, and the training objectives for both DQN and PPO.
4. Implementation details  show technical rigor and awareness of RL stability challenges.
5. The approach is relevant beyond the legal domain — it provides a reusable framework for cost-sensitive orchestration in multi-agent LLM systems.

**Weaknesses:**

1. The proposed architecture assumes access to frozen, deterministic LLM APIs. In practical systems, model drift or context variability may affect reward consistency — this limitation could be discussed more explicitly
2. The approach focuses on discrete action orchestration; it might struggle when actions require parameterized control
3. The impact of goal-conditioning curriculum on convergence speed and generalization is described qualitatively but not visualized

**Questions:**

1. How stable is the learned orchestration policy when deployed on unseen contract types (e.g., NDAs, employment agreements) or different LLM APIs? Would retraining be necessary, or could transfer learning suffice?
2. The reward function balances quality and token cost linearly. Did the authors experiment with nonlinear or adaptive weighting, such as diminishing returns beyond a certain quality threshold?
3. The conclusion mentions possible extension to “select between LLMs per agent.” Could this be modeled as a hierarchical RL setup where higher-level policies choose models and lower-level ones select prompts?

---

### Official Review · Reviewer_LPRh · 2025-10-23

**Soundness:** 1
**Presentation:** 1
**Contribution:** 1
**Rating:** 0
**Confidence:** 5

**Summary:**

This paper presents an orchestrator that learns when to route contract parsing tasks between specialist LLM agents to minimize cost-quality tradeoff. Authors experiment on CUAD, a NLP dataset for contract review + a small proprietary set. Authors experiments claim quality gains at similar cost, with token savings.

**Strengths:**

The paper tackles a practical problem in legal review, with an attempt to reign in cost and token use while maintaining quality in a RL framework. Authors cite several RL families as comparisons and clearly define problem statement in extracting parties, effective date, governing law with a schema.

**Weaknesses:**

1. Evidence is thin and under specified. Metrics are described at a high level, experiments are with weak baselines, and there are no details on prompts, validation studies, human alignment/labels.
2. Writing is very terse, with 2 pages devoted to RL families at a high level without added context towards the problem statement. Examples remain high-level without conditioning on their use in the legal NLP world. For example, The RL family table lists robotics, Atari, one-shot decision as examples. Also the quality of tables is poor, with overlapping columns.
3. The problem is incremental, and focuses on extraction of several terms in a legal contract. Novelty is focused on the reward shaping and goal vector but discussion is limited in this respect.
4. Ablations should have experimental results, not high-level claims, e.g. removing party resolver hurts most.
5. Several references have incorrect author lists and venues.

**Questions:**

1. Can you elaborate on the prompts you used for extraction.
2. What results did you achieve on your ablations?
3. How do agent frameworks compete with your approach, i.e. smolagents, lang graph, etc.

**Details Of Ethics Concerns:**

Highly LLM generated. The writing is terse, high-level. Several citations are fabricated with fake author lists. This should not be a submission and the authors should not be reviewing for ICLR if they are.

---

### Official Review · Reviewer_X7XK · 2025-10-31

**Soundness:** 2
**Presentation:** 2
**Contribution:** 1
**Rating:** 2
**Confidence:** 3

**Summary:**

The paper presents an RL orchestration framework for legal metadata extraction. The goal is to extract metadata fields from a corpus of contracts. The task is formulated as a sequential decision-making problem where the goal is to balance extraction agents with different costs. Empirical evaluation shows favorable performance for the efficiency-performance Pareto front.

**Strengths:**

* Addresses a well-motivated applied problem.
* Provides helpful intuition and reasoning for algorithm design decisions.

**Weaknesses:**

* Despite the focus on empirical results, analysis code doesn’t seem to be attached, and the source of one dataset is unclear (“an enterprise dataset of telecom supplier contracts spanning 15 years” - L316). This makes it very challenging to validate results and build upon them.
* Significance of contributions beyond the specific legal application is unclear.

**Questions:**

* How does the proposed approach compare to existing baselines studied in the literature?
* Is it possible to highlight new insights gained in this work that may apply in other domains?

---

### Official Review · Reviewer_xqe6 · 2025-11-01

**Soundness:** 2
**Presentation:** 3
**Contribution:** 1
**Rating:** 2
**Confidence:** 4

**Summary:**

This paper formulates contract metadata extraction as a sequential decision-making problem and proposes a reinforcement-learning (RL) orchestration framework for coordinating multiple specialist extraction agents. Each “agent” combines an LLM prompt with regex or heuristic post-processing (e.g., for parties, governing law, or dates). The orchestrator learns, via goal-conditioned DQN and PPO, when to call which agent and when to stop, optimizing a reward that balances field-level extraction quality and token usage.

Experiments are conducted on CUAD and a proprietary enterprise-contract corpus. Reported results show (i) a ~2 % improvement in average quality on CUAD for a quality-oriented PPO policy relative to a baseline with rigid orchestration, (ii) ~35 % token savings for an efficiency-oriented policy, and (iii) a Multi-Pass workflow (lightweight first pass for easy fields + PPO policy on complex fields) that maintains baseline quality while reducing cost. The paper argues that RL-based orchestration can help enable scalable, efficient legal document metadata extraction.

**Strengths:**

(1) Clear problem motivation and practical relevance. \
The paper targets a genuine deployment challenge in legal-contract analysis — extracting many metadata fields from long, heterogeneous documents under cost constraints. The emphasis on managing token usage and balancing efficiency with quality is well-motivated for large-scale applications.

(2) Transparent and technically sound setup. \
The RL framework is well defined, with clear notation, reward formulation, and training details for both DQN and PPO. Goal-conditioning and action masking show awareness of standard RL stability techniques.

(3) Empirical exploration of efficiency–quality trade-offs. \
The experiments on CUAD and enterprise data explore how multi-step orchestration can adjust quality–cost behavior and whether reinforcement learning helps optimize these trade-offs. The study provides a useful initial look at cost-aware orchestration in document extraction, even if stronger baselines and deeper analysis would be needed for firm conclusions.

(4) Awareness of limitations. \
The authors acknowledge issues such as reward-shaping bias, dataset sampling bias, and latency from multi-call orchestration. Expanding the discussion and conclusion to reflect more deeply on these challenges and broader implications could make the paper’s insights more impactful.

**Weaknesses:**

(1) Weak and mismatched baselines that do not isolate the contribution of reinforcement learning. \
The current baseline design makes it difficult to determine how much of the reported improvement stems from reinforcement learning versus simply allowing multiple LLM calls. The single-pass LLM baseline is not cost- or call-matched, since the proposed RL approach performs several API calls (one per agent/action). This makes any advantage ambiguous: improvements could arise from having more opportunities to refine extractions rather than from learned sequential decision-making. A fairer comparison would hold the compute budget constant—for example, by allowing a unified instruction-tuned LLM to perform the same number of self-reflection or refinement rounds (e.g., ReAct / Reflexion-style) under identical token limits. Conversely, the static-order scheduler baseline roughly matches the multi-call structure but lacks any adaptivity; it cannot test whether temporal credit assignment or document-specific decision-making actually drives the gains. Both baselines therefore bracket the true problem space without isolating the effect of learning sequential policies. Introducing stronger adaptive, non-RL baselines—such as contextual bandits, self-reflective or budget-controlled LLMs, or confidence-threshold early-exit controllers—would make it much clearer whether the proposed reinforcement-learning framework adds value beyond cost-matched prompting.

(2) Need for stronger, cost-controlled adaptive LLM baselines to test cost–quality trade-offs. \
Beyond single-pass and static baselines, a natural family of comparisons would explicitly trade off quality versus cost by controlling per-call budgets and allowing bounded iterative self-correction. For instance, a unified LLM could perform up to $K$ refinement rounds (reflection or verification) with `max_new_tokens` and reasoning depth as adjustable budget knobs, set so that the average number of calls per document matches the RL policy’s trajectory length. Intuitively, such token-budgeted or multi-pass baselines might already recover much of the observed cost–quality frontier: for easier, local fields (e.g., dates, parties), extra reasoning tokens or a few self-refinement steps yield predictable, monotonic quality gains, and simple token-budget control can approximate the same trade-offs that RL discovers. Indeed, the paper’s own Multi-Pass results already hint that much of the benefit may come from inexpensive first-pass heuristics plus limited refinement, rather than from the RL objective itself. The added value of RL should appear primarily in conditional, per-document adaptivity—skipping expensive agents when confidence is high or allocating budget selectively for hard clauses. Including these non-RL baselines would show whether the RL approach truly learns temporal credit assignment or simply automates cost tuning already achievable with explicit token limits.

(3) Use of hand-engineered “agent” actions could be better justified relative to alternatives. \
The action space consists of calls to specialist extractors that combine LLM prompts with regex and heuristic post-processing (e.g., `PartiesAgent`, `DatesAgent`, `GoverningLawAgent`). This is an interesting and pragmatic way to define discrete actions for the orchestration problem, since these agents capture heterogeneous cost–quality trade-offs and make RL tractable. However, the paper would benefit from a clearer justification of why this particular atomization of the action space is preferable to simpler alternatives—such as formulating contract extraction as a unified, multi-label SQuAD-style span-prediction task (as in CUAD’s original formulation) or using open-form prompting with iterative self-correction. In particular, since the paper reports results on CUAD, it would be informative to compare against a modern instruction-tuned LLM evaluated directly on the CUAD span-prediction format, to assess whether the RL orchestration meaningfully improves over a strong unified baseline. Clarifying the advantages of the “specialist-agent” decomposition (e.g., modularity, interpretability, or ease of cost control) would help readers understand the motivation for this design.

(4) Limited and surface-level reward signal. \
The proposed quality reward relies on token- and string-overlap metrics (token F1, format checks, Jaccard / Levenshtein similarity), which measure syntactic rather than semantic correctness. This may suffice for straightforward fields (dates, amounts) but provides weak feedback for semantically rich clauses (e.g., unlimited license, covenant not to sue), where surface overlap often fails to capture correctness. Without a more semantic reward (e.g., embedding-based similarity, entailment scoring, or clause-level classification accuracy), the learned policies risk optimizing for superficial string similarity rather than genuine metadata accuracy or completeness. This limitation also undermines the interpretation of the “quality” improvements reported in Table 2, since the metric itself may not reflect substantive gains.

(5) Insufficient detail on state representation and feature design. \
The state representation is described as encoding (i) which fields are filled, (ii) document features (e.g., amendment type, financial-contract flag, length), and (iii) cumulative token usage, yet the paper provides no description of how these features are represented—whether as symbolic indicators, scalar summaries, or language embeddings. Based on the task description, one might expect the state to already incorporate text embeddings, but the conclusion’s reference to “future directions … richer state representations (e.g., LegalBERT embeddings)” suggests otherwise. This design choice has important implications for generalization and scalability: a state vector with manually chosen numeric features may not transfer to other domains or contract schemas. A clearer account of how the current state vector is encoded, updated, and normalized—and how it interacts with the goal-conditioning vector $g$ would be helpful.

(6) Ambiguity in the Pareto-frontier interpretation and incomplete sweep over trade-off parameters. \
The discussion of the Pareto frontier (Figure 2) remains unclear. As written, only the efficiency-oriented and multi-pass policies are said to approach the frontier, but if the frontier is defined as the set of non-dominated (quality, token) trade-offs produced by sweeping $(\lambda, \mu)$, then the high-$\lambda$ quality-oriented settings should also lie on that boundary at the high-cost, high-quality end. Without showing the full sweep or the corresponding convex-hull / non-dominated set, it is difficult to determine whether “frontier” is being used informally to mean efficient-cost points or whether those quality-focused policies are actually dominated. A more complete sweep over $(\lambda, \mu)$ and explicit visualization of the convex hull would make the trade-off analysis much clearer. At present, the quality-oriented settings appear to yield little additional accuracy for their added cost, suggesting diminishing returns, mis-scaled rewards, or optimization saturation. Clarifying whether this plateau reflects model capacity limits, reward design, or training instability would make the interpretation more convincing.

(7) Ambiguity and limited interpretability of the reported results. \
Table 2 does not specify which baseline configuration is used for comparison, nor whether all methods were run under comparable token budgets. Moreover, results are reported as single point estimates without confidence intervals or standard deviations. Given that the baseline and quality-oriented PPO scores are nearly identical (0.916 vs. 0.932) while the efficiency-oriented score is substantially lower (0.626), it is difficult to judge whether these differences would hold across seeds or random splits. The paper characterizes this 30-point quality drop as “modest,” but the magnitude appears quite large without additional context. Reporting mean $\pm$ variance across seeds and including qualitative examples of model outputs would help readers assess whether the ~35% token reduction in the efficiency-oriented policy corresponds to an acceptable quality trade-off or visibly degraded extractions. Such evidence is necessary for the results to be interpreted with confidence.

(8) Limited methodological novelty. \
Conceptually, the paper primarily applies established RL algorithms (goal-conditioned DQN and PPO) to an existing extraction task (CUAD) and an enterprise contract corpus, rather than introducing new algorithmic ideas or theoretical insights. Its main contribution is an empirical case study showing that cost-aware RL orchestration can manage quality–cost trade-offs in LLM-based information extraction. This is a reasonable and practically motivated direction, but the technical novelty is limited, and the current experiments fall short of demonstrating a clear advantage over strong adaptive non-RL baselines.

**Questions:**

1. Based on the description of “rigid orchestration” in the abstract, it appears that the Table 2 “Baseline” corresponds to the static scheduler. Could the authors confirm this and describe exactly how this baseline operates? In particular, does the static schedule call each agent once and then stop, or can it call the same agent multiple times as long as the order is fixed? How is the total number of calls controlled in this configuration?

2. The 2% improvement in average quality and 35% reduction in token usage reported in Table 2 are interesting, but no confidence intervals or variance estimates are provided. Have the authors assessed whether these differences are statistically significant? Running the experiments across multiple random seeds or reporting standard deviations would help establish the statistical validity of these results.

3. Could the authors include qualitative examples illustrating what kinds of extraction errors or degradations correspond to the reported F1 drop for the efficiency-oriented policy in Table 2? This would help readers understand whether the quality reduction represents acceptable variation or visibly poorer metadata extraction. At present, the paper describes this as a “modest” trade-off, yet the reported average quality (F1-style) decreases from roughly 0.93 to 0.63, which seems substantial without additional context.

4. How was the Pareto frontier in Figure 2 computed—formally (e.g., convex hull of points from a $(\lambda, \mu)$ sweep) or qualitatively based on a small number of settings? Clarifying this would help interpret which policies are actually Pareto-efficient and how the “efficiency-oriented” and “quality-oriented” variants relate along the trade-off curve.

5. Based on the task description, one might expect the state representation to already incorporate text embeddings (e.g., from LegalBERT), but the conclusion states that “future directions include richer state representations (e.g., LegalBERT embeddings).” Could the authors clarify how the state vector is represented in the current implementation—specifically, how each component (which fields are filled, document features such as amendment type or length, and cumulative token usage) is encoded? Are these represented as symbolic one-hot indicators, continuous scalar summaries, or learned embeddings? Additionally, how is the goal-conditioning vector $g = [\gamma_g, \lambda, \mu]$ incorporated into the policy and value networks—is it concatenated directly to the state features or integrated through another conditioning mechanism? Clarifying these details would help readers understand how the current RL policy processes document-level context and how it differs from the embedding-based approach mentioned for future work.

6. The limitations mention that “reward shaping can bias behavior.” Could the authors elaborate on what types of bias were observed in practice (e.g., premature stopping, over-penalization of costly agents)? How sensitive are the learned policies to the relative magnitudes of the shaping terms (entropy bonus, patience penalty, invalid-action penalty)? A brief ablation on shaping weights would clarify how robust the policy is to these design choices.

---

### Note · Authors · 2025-11-30

**Comment:**

Thank you very much for your invaluable review and comments on my submitted paper. I truly appreciate the time and effort you put into the feedback. However, due to several work deadlines at the moment, I am unable to revise the paper as required and unfortunately must withdraw it at this time. I sincerely apologize for any inconvenience this may cause.

**Withdrawal Confirmation:**

I have read and agree with the venue's withdrawal policy on behalf of myself and my co-authors.